# Defactinib inhibits PYK2 phosphorylation of IRF5 and reduces intestinal inflammation

Grigory Ryzhakov[1,4,5], Hannah Almuttaqi[1,5], Alastair L. Corbin [1], Dorothée L. Berthold [1], Tariq Khoyratty [1], Hayley L. Eames[1], Samuel Bullers[1], Claire Pearson[1], Zhichao Ai[1], Kristina Zec[1], Sarah Bonham[2], Roman Fischer [2], Luke Jostins-Dean [1], Simon P. L. Travis [3], Benedikt M. Kessler [2] & Irina A. Udalova [1✉]

Interferon regulating factor 5 (IRF5) is a multifunctional regulator of immune responses, and has a key pathogenic function in gut inflammation, but how IRF5 is modulated is still unclear. Having performed a kinase inhibitor library screening in macrophages, here we identify protein-tyrosine kinase 2-beta (PTK2B/PYK2) as a putative IRF5 kinase. PYK2-deficient macrophages display impaired endogenous IRF5 activation, leading to reduction of inflammatory gene expression. Meanwhile, a PYK2 inhibitor, defactinib, has a similar effect on IRF5 activation in vitro, and induces a transcriptomic signature in macrophages similar to that caused by IRF5 deficiency. Finally, defactinib reduces pro-inflammatory cytokines in human colon biopsies from patients with ulcerative colitis, as well as in a mouse colitis model. Our results thus implicate a function of PYK2 in regulating the inflammatory response in the gut via the IRF5 innate sensing pathway, thereby opening opportunities for related therapeutic interventions for inflammatory bowel diseases and other inflammatory conditions.

[1] University of Oxford, Kennedy Institute of Rheumatology, Oxford, United Kingdom. [2] Target Discovery Institute, Nuffield Department of Medicine, Centre for Medicines Discovery, Chinese Academy of Medical Sciences Oxford Institute, University of Oxford, Oxford, United Kingdom. [3] Translational Gastroenterology Unit, NIHR Oxford Biomedical Research Centre, Oxford University Hospitals NHS Foundation Trust, John Radcliffe Hospital, Oxford, United Kingdom. [4] Present address: Novartis Institutes for BioMedical Research, Novartis Pharma AG, Novartis Campus, Basel, Switzerland. [5] These authors contributed equally: Grigory Ryzhakov, Hannah Almuttaqi. ✉email: irina.udalova@kennedy.ox.ac.uk

nflammatory bowel disease (IBD) is a group of inflammatory disorders of the gastrointestinal tract caused by a complex combination of genetic and environmental factors. Genome-wide association studies (GWAS) revealed over 200 genetic loci linked to inflammatory bowel disease (IBD)[1,2]. A recent single-cell transcriptomic analysis of colon biopsies from patients with ulcerative colitis (UC) provided a framework for linking GWAS risk loci with specific cell types and functional pathways and helped to nominate causal genes across GWAS loci[3], amongst them Interferon regulatory factor 5 (IRF5). IRF5 is a multi-functional regulator of immune responses[4–6]. The IRF5 risk variant has consistent effects across monocytes and macrophage conditions, but also impacts gene expression and splicing across a wide range of other immune cells and tissues[7].

Recent studies using IRF5-deficient mice have established a critical role of this transcription factor in the pathogenesis of mouse models of colitis[8,9]. For instance, mice with a global or mononuclear phagocyte (MNP)-specific loss of IRF5 were protected from $Hh + \alpha IL10R$ colitis[9]. Thus, IRF5 is an attractive target for the treatment of IBD. However, the molecular mechanisms of IRF5 activation are still debated. IRF5 is proposed to exert its molecular function via a cascade of events involving its phosphorylation, ubiquitination, dimerisation, nuclear translocation and selective binding to its target genes to enable their expression[10]. Several kinases including TBK1, RIP2, IKKε, IRAK4, TAK1 and IKKβ have been proposed to phosphorylate and activate IRF5[11–17], while IKKα inhibits IRF5[18]. Lyn, a Src family kinase has been shown negatively to regulate IRF5 in the TLR-MyD88 pathway in a kinase-independent manner via direct binding to IRF5[19]. Given the importance of phosphorylation on IRF5 activation, we set out to identify novel IRF5 kinases as potential targets for specific control of IRF5-driven inflammatory conditions. In this study, we used reporter-based screening of a kinase inhibitor library to identify PTK2B/PYK2 as an upstream regulator of IRF5.

PTK2B/PYK2 is a non-receptor protein-tyrosine kinase and plays an important role in inflammatory diseases. For example, PYK2 is essential for cell migration in vitro, regulates airway inflammation, Th2 cytokine secretion and airway hyperresponsive in a mouse model of asthma[20]. PTK2B/PYK2 function in macrophages has been linked to complement-mediated phagocytosis[21], membrane ruffling and migration towards sites of inflammation and the secretion of some pro-inflammatory cytokines[22,23]. PYK2 is also another nominated causal gene for UC[3].

In this study, we identify PTK2B/PYK2 as an upstream regulator of IRF5 based on a systematic screen of kinase inhibitors in macrophages. CRISPR-Cas9-mediated knockout of PYK2 in murine RAW264.7 macrophages impairs LPS-induced IRF5 activation and IRF5-dependent expression of pro-inflammatory cytokines. Furthermore, treatment with a selective PYK2 inhibitor defactinib abrogates the induction of macrophage pro-inflammatory transcriptome to a similar extent seen in IRF5 knockout cells. Defactinib reduces pathology in murine model colitis, and cytokine expression in colonic mucosal biopsies collected from patients with active ulcerative colitis. Taken together, this study demonstrates a major role for PYK2 as a key regulator of IRF5 activation, macrophage inflammatory response, and intestinal pathology and may facilitate the development of new therapeutics.

## Results

**Small molecule library screen and in vitro validation of candidate IRF5 kinases.** We have previously established an in vitro reporter system for measuring IRF5-dependent transcription based on the TNF (IRF5-dependent gene)-promoter-driven luciferase construct, which contains a number of interferon-stimulated response elements (ISREs)[24]. This reporter system was used to screen a library of small molecules[25], for which inhibitory properties against 221 protein kinases in the protein kinase inhibitors screen (PKIS) have been established (Supplementary Fig. 1a–c). After the first screen in RAW264.7 macrophages and two rounds of rescreening using different cell types and three different inhibitor concentrations, we composed the final list of 34 candidate IRF5 kinases, among which TBK1, IKKe and IRAK4 were previously proposed to target IRF5[12,16,26] (Fig. 1a and Supplementary Data 1).

For further functional validation, we selected poorly explored proteins or those with known links to inflammatory processes – PYK2, HIPK4, ARK5, CLK2, MARK3, JNK2 and MST1. As controls, we included into these assays RIP2 kinase, the known intermediate in the IRF5-dependent innate immune signalling pathways[11]. When we overexpressed the kinases with IRF5 and TNF-luciferase reporter in HEK-293 TLR4/CD14/MD-2 cells, we found that overexpression of PYK2, JNK2 or MARK3, boosted IRF5-dependent TNF-reporter activation (Fig. 1b). Similar to the known IRF5 binding partners RIP2 and RelA[11,27,28], PYK2 could strongly bind IRF5 in co-immunoprecipitation assays, while HIPK4, ARK5 and JNK2 showed only a weak association with IRF5 (Fig. 1c). We further tested the ability of these kinases to phosphorylate overexpressed IRF5 in 293 ET cell lysates (Supplementary Fig. 1d). We were able to detect phosphorylated IRF5 in the presence of HIPK4, CLK2, JNK2, MST1, PYK2 and RIP2 as a positive control (Fig. 1d). Lastly, we examined the evidence of genetic association of the selected kinases with IBD and found that PYK2 was the only known genetic risk factor[29] (Supplementary Fig. 1e). Based on observed functional interactions with IRF5 and genetic association with IBD, PYK2 was singled out for further investigation.

To map the regions necessary for IRF5-PYK2 interaction, we used expression plasmids that encode truncated mutants of IRF5 or PYK2 (Fig. 1e, f). IRF5 deletion mutants were transfected in HEKTLR4 cells with WT-PYK2 and lysates were subjected to a Myc immunoprecipitation. Constructs lacking the C-terminal serine-rich region (SRR) including IRF5-N220, IRF5-N395 and IRF5-N130 failed to interact with PYK2 (Fig. 1e, g), suggesting that the SRR in IRF5 is essential for its interaction with PYK2. We then carried out a reciprocal Co-IP with PYK2 deletion mutants and WT-IRF5 to identify the domain important for PYK2 interaction with IRF5. The removal of PYK2 central kinase domain in truncation mutant PYK2 1-415 and PYK2 681-1009 resulted in loss of binding to WT-IRF5 (Fig. 1f, h). Thus, IRF5 SRR and PYK2 kinase domain are important for IRF5-PYK2 interactions.

**PYK2 regulates IRF5 activation and IRF5-mediated transcription.** In RAW264.7 macrophages we could detect PYK2 binding to IRF5 at the endogenous level (Fig. 2a). In line with previous studies, we also found LPS-induced PYK2 phosphorylation on Y402 (Fig. 2b)[30,31]. To investigate the kinase's role in the TLR4/IRF5 signalling axis, we generated IRF5- and PYK2-deficient mouse RAW264.7 macrophages using a CRISPR-Cas9 approach (Supplementary Fig. 2a). First, we explored the impact of PYK2 deficiency on IRF5-dependent signalling by transfecting wild type (WT) and PYK2-deficient RAW264.7 macrophages with IRF5-expressing and the TNF-promoter-driven luciferase plasmid. We observed a marked reduction of the LPS-induced reporter activity in IRF5-expressing cells lacking PYK2 (Fig. 2c). When we expressed recombinant PYK2 in PYK2-deficient cells, we achieved a partial reconstitution of the PYK2 levels in

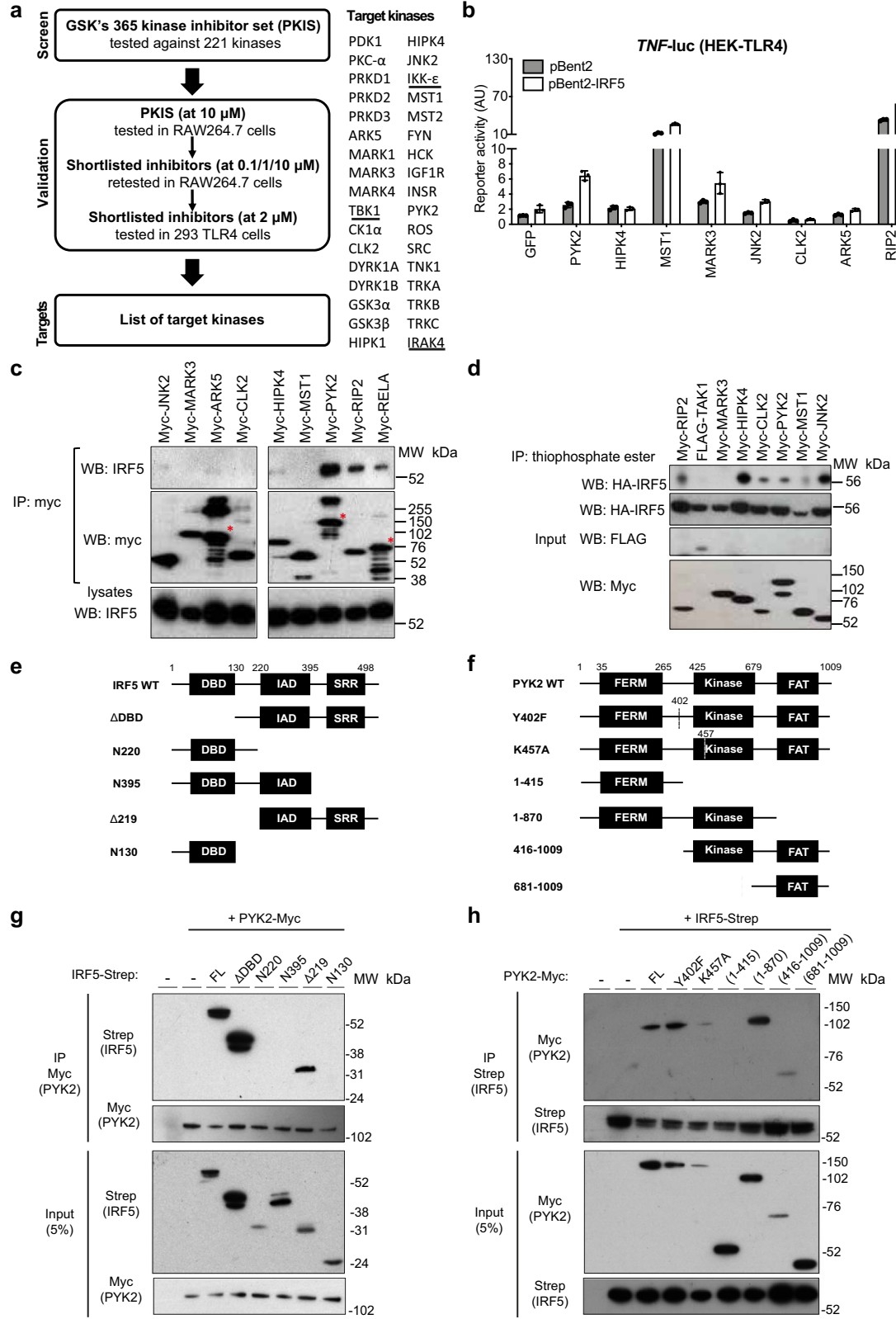

RAW264.7 macrophages and a partial restoration of the reporter activity (Supplementary Fig. 2b, c). Next, we examined if PYK2 deficiency would directly impact IRF5 activation and function by measuring IRF5 recruitment to its target gene promoter and enhancer regions using chromatin immunoprecipitation (ChIP) assay[4,24]. IRF5 recruitment to *Il6*, *Il1a* and *Tnf* gene promoters was impaired in PYK2 knockout cells (Fig. 2d and Supplementary

Fig. 2d). Consequently, we observed attenuated recruitment of RNA polymerase II at the same promoters indicating reduced gene transcription (Fig. 2d and Supplementary Fig. 2d). We also detected a reduction in mRNA induction of these cytokines, as well as chemokines *Ccl4*, *Ccl5*, by LPS in PYK2-deficient cells, comparable to or even stronger than in IRF5 knockout cells (Fig. 2e and Supplementary Fig. 2e). Conversely, LPS-induced IL-

**Fig. 1 Small molecule library screen and in vitro validation identifies PYK2 as a putative IRF5 kinase. a** The screening workflow showing the initial large screening in RAW264.7 cells, and the subsequent screens in RAW264.7 and 293 TLR4 cells. The top inhibitors were shortlisted based on their efficacy towards IRF5 reporter and low toxicity. Based on the known activities of these molecules against 221 kinases in the PKIS set, 34 kinases affected by the top inhibitors were shortlisted. The underlined kinases were previously proposed to target IRF5. **b** Impact of the kinases on the IRF5 reporter activity. Luciferase activities were measured in cells co-expressing IRF5 (or empty plasmid control, pBent2), TNF-luc reporter and one of the shortlisted kinases. Reporter activity was calculated as firefly luciferase activity normalised against constitutively expressed Renilla luciferase units and is shown as compared to the values in cells not expressing any kinase. AU arbitrary units of luminescence. Data presented as mean values ± SD for $n = 3$ independent experiments. **c** Binding of IRF5 to the shortlisted kinases. Myc-tagged kinases and HA-tagged IRF5 were co-expressed in 293 ET cells. Cell lysates were subjected to immunoprecipitation (IP) using anti-myc antibody and levels of kinases and IRF5 in the IP eluates and proteins were determined by western blot. Asterisks indicates the expected molecular weight. **d** In vitro kinase assays of 293 ET cells co-transfected with HA-IRF5 and myc- or flag-tagged kinases. Proteins in the pull-downs and lysates were detected by Western blotting using antibodies against HA- (IRF5) and myc- and FLAG- (kinases). **e** Schematic of IRF5 truncation mutants. **f** Schematic of PYK2 truncation mutants. **g**, **h** HEKTLR4 cells were co-transfected with Myc-PYK2 and Strep-tagged IRF5 truncation mutants and subjected to co-immunoprecipitation and western blot analysis. Representative blots from three independent experiments are shown. Source data are provided as a Source Data file.

10 induction was increased in PYK2 knockout (Supplementary Fig. 2e), similarly to our previous findings in IRF5 knockout cells[4].

To validate our observations in primary cells, we utilised immortalised myeloid progenitor HoxB8 cells[32], which differentiated into non-proliferating mature macrophages in the presence of GM-CSF (Supplementary Fig. 3a). Using the CRISP-Cas9 approach, we generated stable knockout of IRF5 and PYK2 in these cells and validated their absence by western blot analysis (Supplementary Fig. 3b). After 5 days of ex vivo differentiation in the presence of GM-CSF, HoxB8 progenitors deficient in IRF5 or PYK2 gave rise to mature macrophages, comparable to the wild-type cells, but the levels of inflammatory cytokine and chemokine production were significantly reduced in HoxB8 macrophages deficient in IRF5 or PYK2 (Fig. 2f). Thus, PYK2 acts as a critical regulator of IRF5 activation and inflammatory response induced by LPS in macrophages.

**PYK2 phosphorylates IRF5**. To characterise potential PYK2 target residues in IRF5, we employed phospho-proteomics. Endogenous IRF5 was immunoprecipitated from the lysates of LPS-stimulated WT and PYK2-deficient RAW264.7 macrophages, and the phospho-peptides were further enriched from the total proteolytic digests. Peptide masses and quantities were analysed by nano ultra-high-pressure liquid chromatography coupled with mass spectrometry (nUPLC-MS/MS). In line with previously published reports[14,15] we identified the Ser-445 IKKβ-dependent site in both WT and PYK2-deficient cells (Fig. 3a and Supplementary Fig. 4a, b). In addition, endogenous IRF5 was phosphorylated at residues Ser-172, Ser-300, Tyr-334 in both WT and PYK2-deficient cells (Fig. 3a and Supplementary Fig. 4a, b). Interestingly, we could only detect Y171 phosphorylation in WT cells (Fig. 3a, b and Supplementary Fig. 4b), while S56 and Y312 residues were modified in PYK2-deficient cells (Fig. 3a and Supplementary Fig. 4a, b), possibly reflecting on modification by other enzymes. In fact, Src tyrosine kinase Lyn was capable of phosphorylating IRF5 at orthologues of Y312 (human Y329) and Y334 (human Y351) sites (Supplementary Fig. 4c) in in vitro co-expression system[19].

We individually mutated these sites Y171 (human Y172), Y312 (human Y329), Y334 (human Y351) as well as the published Y104 site[33] into phenylalanine residues and explored the consequence of these mutations in the above-mentioned reporter and in vitro phosphorylation assays (Supplementary Fig. 1a, d). We observed a reduction in PYK2-dependent phosphorylation of human v2 IRF5 Y172 and Y172/S173 but not other IRF5 mutants (Fig. 3c).

The Y172F mutant of human v2 IRF5 and the double mutant Y172/S173A (Supplementary Fig. 4d) both showed somewhat

diminished ability to activate the TNF-luciferase reporter in the presence of PYK2, whereas the Y104F mutation had no inhibitory effect (Supplementary Fig. 4d). Notably, Y329F and Y351F mutant of human v2 IRF5 also displayed reduced TNF-reporter activity (Supplementary Fig. 4d) and warrant further exploration.

Together, these results indicate that PYK2 contributes to the LPS-induced phosphorylation of IRF5.

**A PYK2 inhibitor Defactinib interferes with IRF5 activation**. Recently, specific inhibitors of PYK2 and a related kinase FAK have been developed[34]. One of them called defactinib (also known as VS-6063 or PF-04554878), with high selectivity to PYK2 and FAK1 and low affinity to kinases outside the family[35], has been successfully used to tackle cancer in a mouse model and is currently being tested in clinical trials[36,37]. Here we explored the effect of defactinib on IRF5 activation and IRF5-dependent gene expression by macrophages. We first established the concentration range of defactinib well tolerated by RAW264.7 macrophages (Supplementary Fig. 5a). At such concentrations (0.3–1 μM), it reduced LPS-induced PYK2 phosphorylation (Supplementary Fig. 5b) and effectively inhibited TNF-reporter activity and gene expression in RAW264.7 macrophages in a dose-dependent manner (Supplementary Fig. 5c, d). Moreover, defactinib inhibited TNF-reporter activity in wt and IRF5-deficient RAW264.7 macrophages in which IRF5 expression was restored via ectopic expression of IRF5, but not in PYK2-deficient cells (Fig. 4a), suggesting that defactinib acts in an IRF5- and PYK2-specific manner. Supporting this notion, defactinib had no further effect on cytokine expression in HoxB8 macrophages deficient in PYK2 (Supplementary Fig. 5e).

Mechanistically, defactinib prevented LPS-induced nuclear translocation of IRF5 but not p65/RelA (Fig. 4b). We observed little effect on the p65/RelA phosphorylation and IkBa degradation in PYK2 KO cells and/or in cells treated with defactinib (Supplementary Fig. 6a), ruling out a major role for NF-κB in PYK2 signalling pathway in these cells. We also examined JNK signalling pathway as a potential off-target effect of defactinib[38] and found no differences in pJNK levels when comparing WT and PYK2 KO cells treated with defactinib (Supplementary Fig. 6a).

Mirroring our PYK2 deficiency data in RAW264.7 macrophages (Fig. 2), LPS-induced IRF5 and RNA polymerase II recruitment to its target promoters was suppressed in primary mouse bone marrow-derived macrophages (BMDMs) treated with defactinib (Fig. 4c). The expression of some pro-inflammatory cytokines and chemokines was also effectively suppressed by defactinib (Fig. 4d and Supplementary Fig. 6c), without affecting cell viability (Supplementary Fig. 6b). Similar results were obtained in macrophages in response to activation of

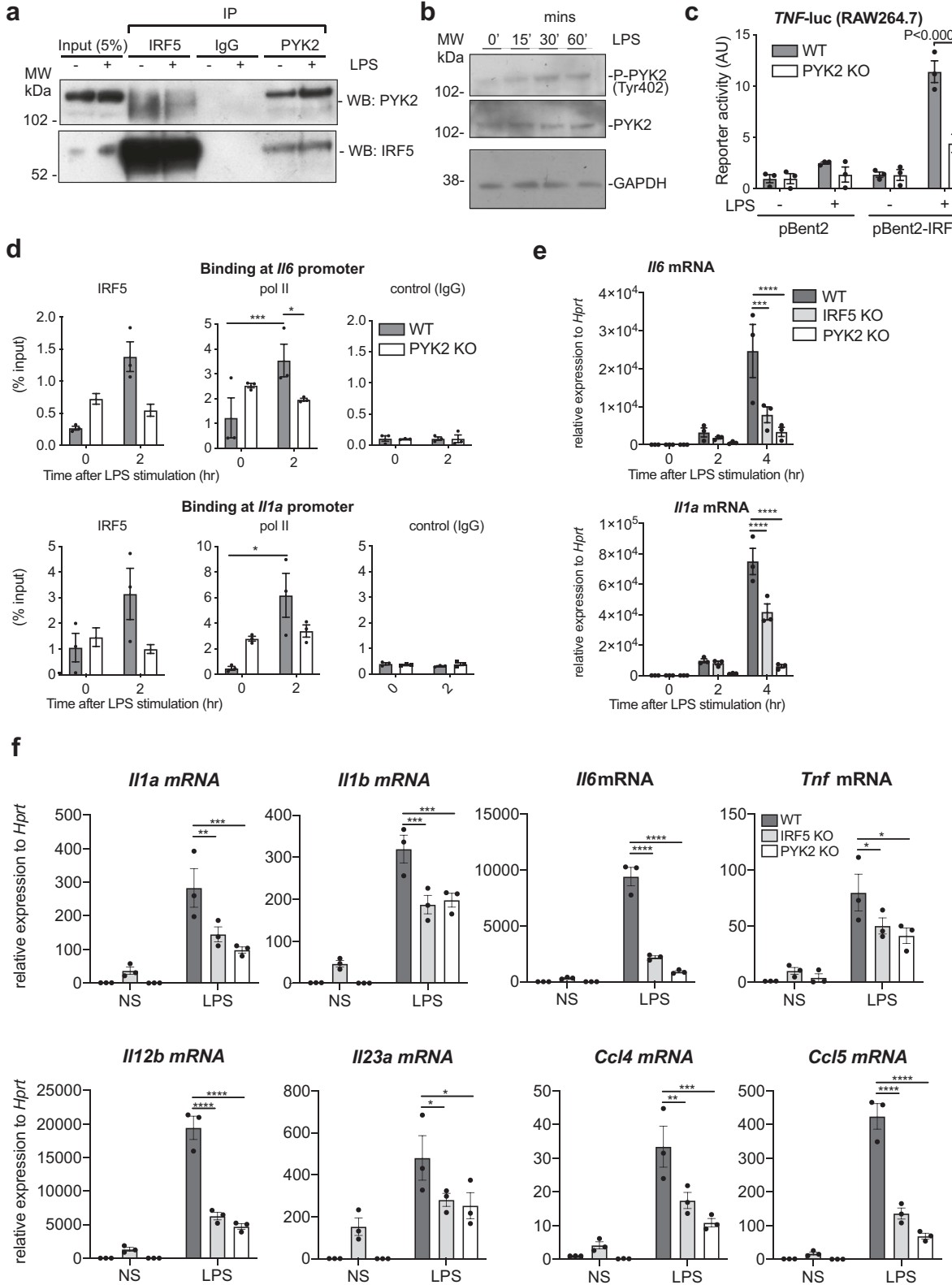

C-type lectin receptor Dectin-1 pathway, in which IRF5 has been shown to play a key role[39], with WGP (dispersible whole glycan particles) (Supplementary Fig. 6d).

**Defactinib treatment phenocopies the effect of IRF5 deficiency on macrophage transcriptome.** To investigate the global impact of PYK2 inhibition on IRF5 target gene expression, we compared LPS-induced transcriptomes in WT and IRF5 KO BMDMs treated with either defactinib or vehicle. Principle component analysis (PCA) of differentially expressed genes (DEGs) ($p < 0.05$) clearly separated WT and IRF5 KO, as well as untreated and LPS-treated sample groups (Fig. 5a). LPS-stimulated WT samples treated with defactinib or vehicle were also clearly separated, with the defactinib-treated samples grouping closely with the IRF5$^{-/-}$.

**Fig. 2 PYK2 interacts with IRF5 and controls IRF5 activation. a** Endogenous co-immunoprecipitation in RAW264.7 macrophages. Cells were stimulated with LPS for 10 min and immunoprecipitated with IRF5, PYK2 or an isotype control antibody. Immunoprecipitates were eluted from IP beads and proteins present in cell lysates (5% inputs) and eluates were detected by immunoblotting with antibodies against IRF5 or PYK2. **b** Immunoblot of LPS-induced PYK2 tyrosine phosphorylation. Blots were probed with antibodies specific for PYK2 phosphorylated on Tyr-402, total PYK2 and GAPDH. Representative blots from three independent experiments are shown for (**a**, **b**). **c** TNF-luc reporter activity in the absence or presence of ectopically expressed IRF5 in wild type and PYK2 KO RAW264.7 cells stimulated with LPS 6 h or left untreated. AU arbitrary units of luminescence. **d** IRF5 and pol II binding to *Il6* and *Il1a* gene promoter in resting or LPS-treated (2 h, 500 ng/ml) wild type or PYK2 KO RAW264.7 cells as measured by the chromatin immunoprecipitation (ChIP) method. A non-specific IgG antibody was used as a negative control for ChIP. **e** *Il6* and *Il1a* mRNA induction in wild type, PYK2 KO or IRF5 KO RAW264.7 cells stimulated with LPS (500 ng/ml) for 0, 2, or 4 h. Gene expression was measured by qPCR. All values in (**c–f**) are shown as mean values ± SEM from $n = 3$ independent experiments. Statistical significance was calculated with two-way ANOVA with Sidak's correction *$P < 0.05$, **$P < 0.01$, ***$P < 0.001$, and ****$P < 0.0001$. **f** Gene expression levels in wild type, PYK2 KO or IRF5 KO HoxB8 macrophages stimulated with LPS (100 ng/ml) 2 h. Gene expression was measured by qPCR. Values shown as mean values ± SEM from $n = 3$ experiments. Comparison by two-way ANOVA (*$P < 0.05$, **$P < 0.01$, ***$P < 0.001$, and ****$P < 0.0001$). Source data are provided as a Source Data file.

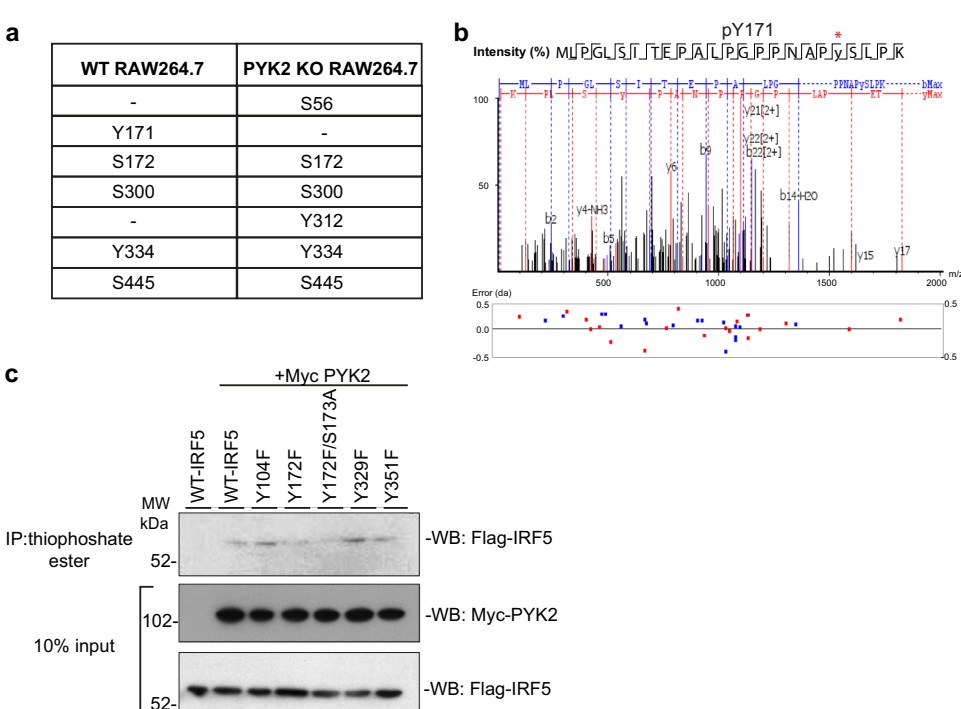

**Fig. 3 PYK2 phosphorylates IRF5. a** Phosphorylation sites identified in LPS-stimulated WT and PYK2 KO RAW264.7 cells. **b** MS/MS spectrum of the IRF5 derived tryptic peptide 152-179 indicating phosphorylation at positions Y171. Fragmentation ions of the b- and y- series are indicated in blue and red, respectively. **c** In vitro kinase assay and immunoblot of IRF5-site-specific tyrosine mutants. HEKTLR4 cells were co-transfected with FLAG-IRF5 tyrosine mutants as indicated and Myc-PYK2. 10% lysate was kept for input and the remaining was used for in vitro kinase reactions. Kinase assays were detected by western blot using antibodies against Flag-(IRF5) and Myc-(PYK2). Representative blots from three independent experiments are shown. Source data are provided as a Source Data file.

Conversely, defactinib had very little effect on IRF5$^{-/-}$ cells stimulated with LPS (Fig. 5a). This was reflected in the number of DEGs: 4,026 for WT and only 217 for IRF5$^{-/-}$ BMDMs treated with defactinib at 2 h of post LPS stimulation (Fig. 5b, e). Gene ontology (GO) analysis for defactinib downregulated genes revealed that they are predominantly pro-inflammatory in nature (e.g. cellular response to interferon-beta, regulation of inflammatory response, cytokine activity, etc). These GO terms were also enriched in LPS-induced genes, and in IRF5 upregulated genes, suggesting that defactinib is highly specific for IRF5 target genes[28] (Fig. 5c). We next investigated the correlation between IRF5 regulated genes and defactinib target genes. The majority of IRF5 upregulated genes were strongly repressed by defactinib and there was a high degree of overlap between IRF5 up- and defactinib down-regulated genes, including *Il1a*, *Il1b*, *Il6*, *Il12a*, *Il12b*, *Il23a*, *Ccl3*, *Ccl4*, etc (Fig. 5d, e). Interestingly, there was also a smaller overlap between IRF5 downregulated genes and

defactinib upregulated genes, suggesting that the actions of IRF5 and defactinib are in direct opposition to each other.

**Defactinib reduces inflammation in mouse Hh/anti-IL10R-model of colitis.** IRF5 activity in mononuclear phagocytes (MNPs) plays a critical role in the pathogenesis of intestinal inflammation and that mice deficient in IRF5 are protected from overblown colitis[8,9]. Here we explored if inhibition of PYK2 would also improve the intestinal immunopathology in a model of *Helicobacter hepaticus* infected and anti-IL-10R monoclonal antibodies administered (Hh+anti-IL10R) colitis, which is characterised by IL-23-dependent intestinal inflammation along with a robust T helper type 1/type 17 (Th1/Th17)-polarised effector T cell response[40] (Supplementary Fig. 7a). As expected, Hh+anti-IL10R-infected mice developed inflammation in the colon after a week. However, intestinal pathology, as well as immune cell

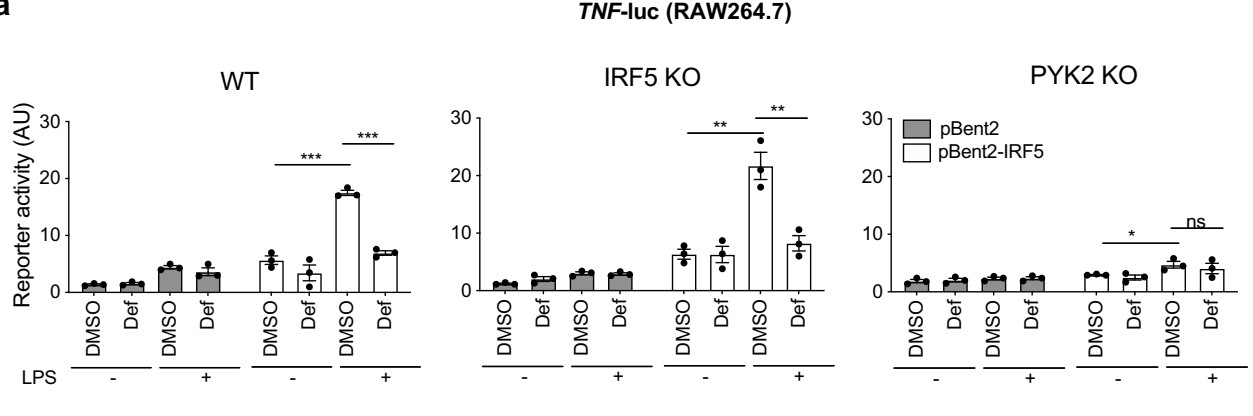

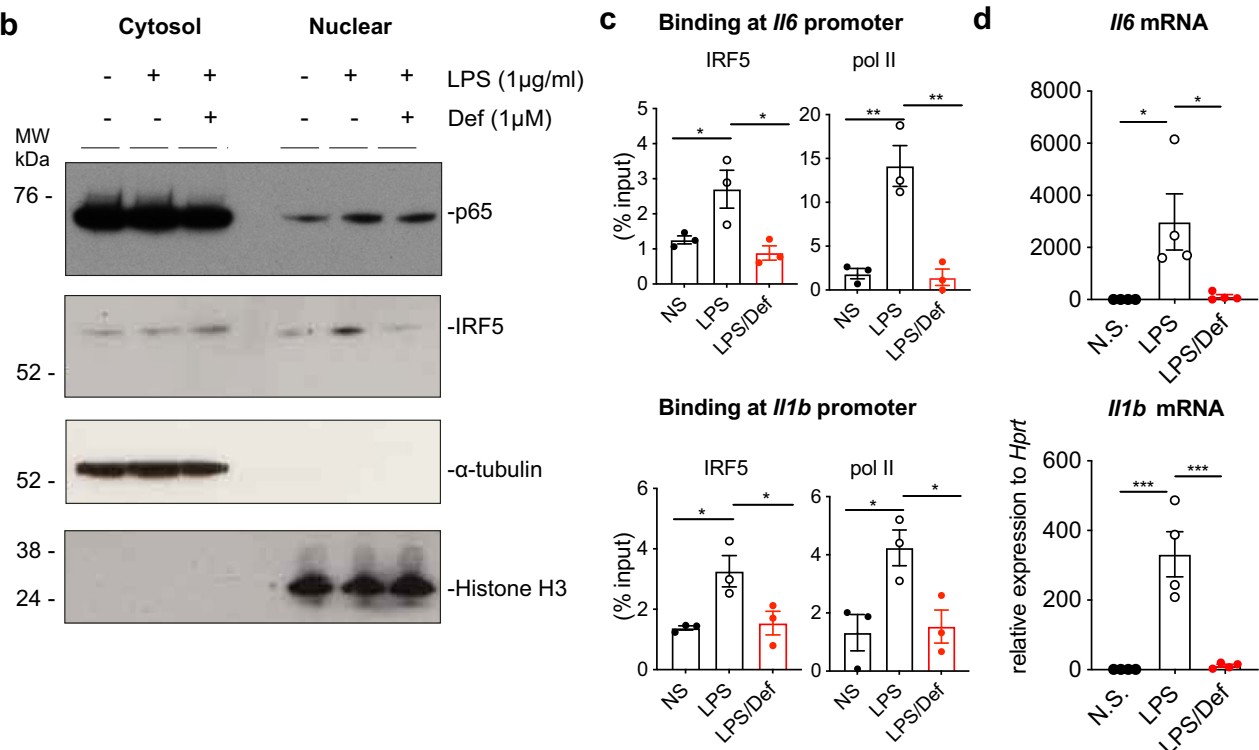

**Fig. 4 A PYK2 inhibitor Defactinib suppresses IRF5 activation. a** TNF-luc reporter activity in the absence or presence of ectopically expressed IRF5 in wild type, IRF5 KO and PYK2 KO RAW264.7 cells pretreated for 1 h with 1 μM of defactinib (or DMSO control) and then stimulated with 1 ug/ml of LPS for 6 h or left untreated. AU arbitrary units of luminescence. Data were shown as means ± SEM from $n = 3$ independent experiments. Comparison by two-way ANOVA with Tukey's correction (*$P < 0.05$, **$p < 0.01$ and ***$P < 0.001$). **b** RAW264.7 cells were fractionated into cytosolic and nuclear extracted following 1 h pretreatment with defactinib (1 μM) and 2 h stimulation with LPS (1 μg/ml). Representative blots from three independent experiments are shown. **c** IRF5 and pol II binding to *Il6* and *Il1b* gene promoters in GM-CSF-differentiated mouse BMDMs pretreated with 3.5 μM defactinib (def) or DMSO control and further stimulated with LPS for 2 h. Chromatin recruitment was analysed by ChIP. Data were normalised against chromatin amount in lysates (and expressed as a percentage of input for each gene) and shown as mean values ± SEM from $n = 3$ independent experiments. Comparison by one-way ANOVA with Tukey's correction (*$P < 0.05$, **$P < 0.01$). **d** *Il6* and *Il1b* expression levels in GM-BMDMs pretreated with 3.5 μM defactinib (def) or DMSO control for 1 h, followed by LPS stimulation for 2 h. Data were shown as means ± SEM for $n = 4$ individual mice. Statistical significance was analysed by one-way ANOVA with Tukey's correction (*$P < 0.05$ and ***$P < 0.001$). Source data are provided as a Source Data file.

infiltrate, and PYK2 activation in colon tissue were reduced in the animals, which received defactinib (Fig. 6a–c and Supplementary Fig. 7b, c). We also observed attenuated induction of pro-inflammatory cytokines and chemokines[41] in the colon of defactinib-treated Hh+anti-IL-10R infected animals (Fig. 6d). In response to defactinib, expression of *Il6, Tnf, Il2b* and *Ccl4* was downregulated in all samples analysed: (1) colon tissue, (2)

leucocyte-enriched colonic cells and (3) isolated colonic mono-cytes/macrophages (Supplementary Fig 7d). Interestingly, the reduction of *Ccl5* expression and an upward trend in *Il10* expression was only observed in isolated monocytes/macro-phages, indicating that the production of these mediators by other cells may mask the effect of IRF5 pathway inhibition in macro-phages (Fig. 6d).

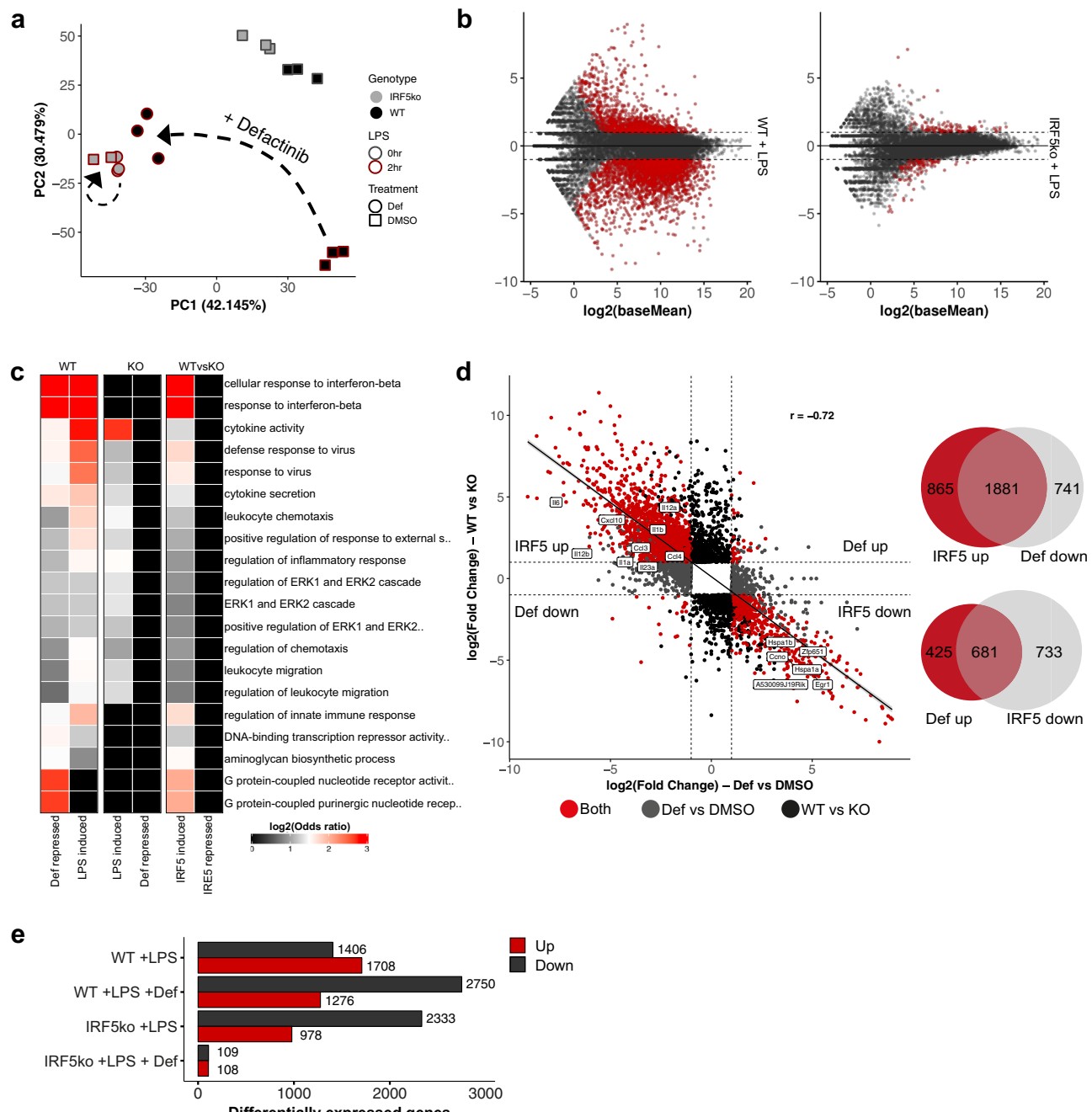

**Fig. 5 Defactinib inhibits IRF5-dependent gene expression. a** PCA analysis of RNA-seq data from WT and Irf5$^{-/-}$ pretreated with 3.5 μM defactinib (def) or DMSO control for 1 h and further stimulated with LPS for 0 or 2 h. **b** MA plots depicting the effect of defactinib on LPS-stimulated BMDMs from WT or IRF5$^{-/-}$ mice. Differentially expressed genes (fold change > 1 and padj < 0.05) are highlighted in red. **c** GO enrichment analysis for differentially expressed genes (as in **b**). **d** Correlation analysis of IRF5 and defactinib regulated genes after 2 h LPS stimulation. Red indicates genes are differentially expressed (significance as in **b**) in both comparisons, genes regulated by IRF5 only (black), genes regulated by defactinib only (grey). Venn diagrams demonstrate the overlap between IRF5 and defactinib regulated genes. **e** Number of DE genes from RNA-seq.

**Defactinib inhibits the production of inflammatory mediators by human monocyte-derived macrophages and UC biopsies.** We next investigated whether PYK2 inhibition suppressed IRF5-dependent innate sensing and inflammatory response in human macrophages. Similar to our finding in mouse BMDMs, we saw robust inhibition of LPS-induced expression and production of IRF5-dependent cytokines in human monocyte-derived macrophages treated with defactinib at 0.5–5 μM concentrations, which did not affect cell viability (Fig. 7a, b and Supplementary Fig. 8a). Next, we tested the impact of PYK2 inhibition in biopsies derived

from the colonic mucosa of patients with active ulcerative colitis by measuring cytokine production at concentrations not affecting cell viability in these samples (Supplementary Fig. 8b, c). We found elevated cytokine production in biopsies obtained from the sites of active inflammation in comparison to those from the adjacent non-inflamed colon. Incubation with increasing doses of defactinib significantly lowered IL6 and IL12p70, identified as part of a cassette of inflammatory molecules that mark anti-TNF-resistant IBD[42] (Fig. 7c). We also observed a trend towards increased IL-10 production by the defactinib-treated biopsies, but

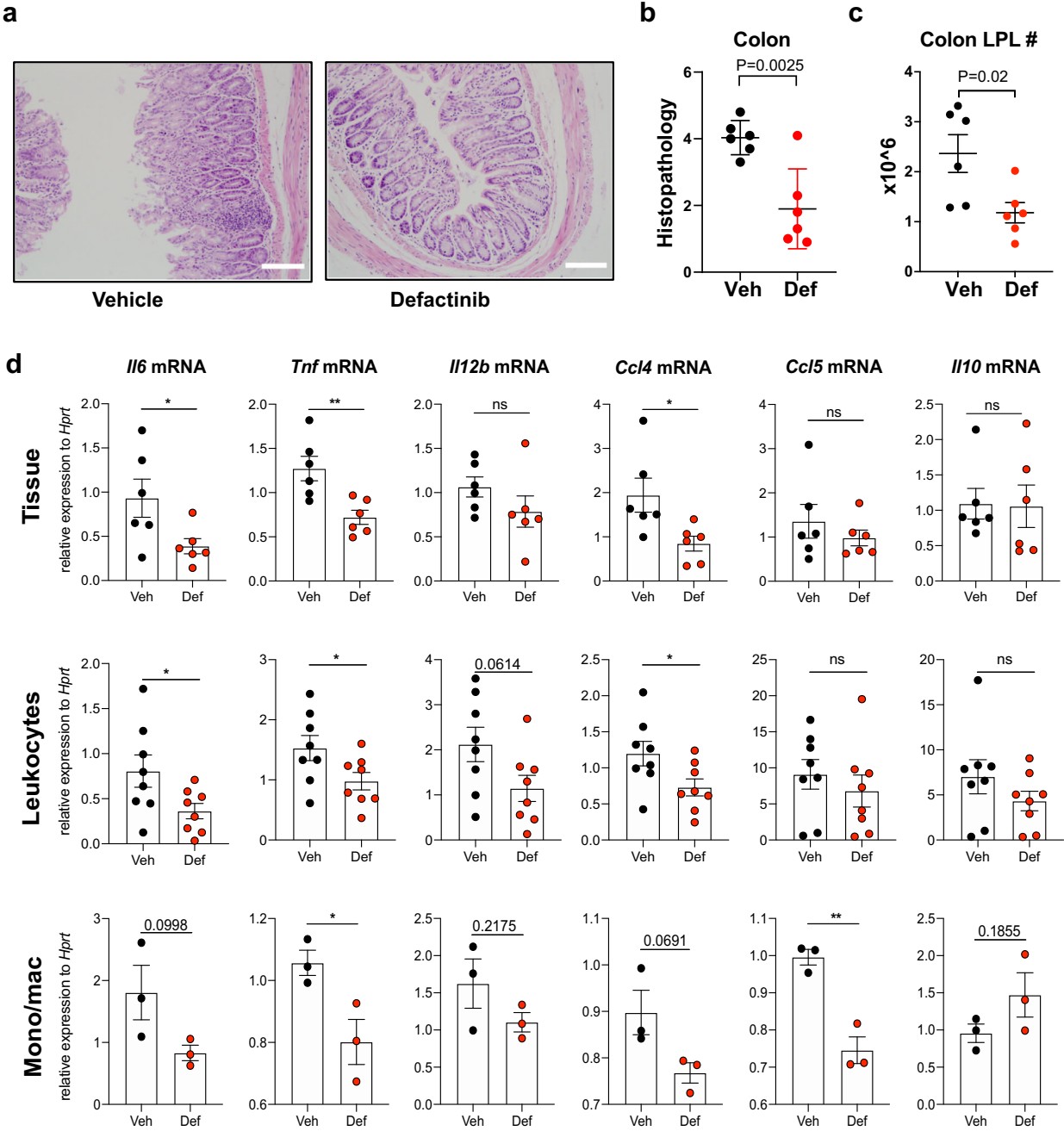

**Fig. 6 Defactinib reduces inflammation in Hh/anti-IL10R-model of murine colitis. a** H&E staining of large intestine tissue sections. Scalebar depicts 100 μM. **b** histology scoring and **c** leucocyte content from *Hh*/anti-IL10R-treated mice, which received either vehicle or defactinib. Data in (**b, c**) shown as mean values ± SEM from n = 6 mice per condition. Statistical significance was calculated by a two-tailed unpaired *t*-test where P = 0.0025 for colon histopathology and P = 0.02 for leucocyte content. **d** Cytokine/chemokine mRNA expression levels in mouse colon tissue, leucocyte-enriched colonic cells, monocytes/macrophages from a vehicle or defactinib Hh/anti-IL10R treated mice. Data were shown as mean values ± SEM from n = 6-8 mice per condition for tissues and leukocytes. Samples were pooled to give n = 3 for monocytes/macrophages. Statistical significance was calculated by a two-tailed unpaired *t*-test (*P < 0.05 and **P < 0.01). Source data are provided as a Source Data file.

production of IL1β appeared to be unaffected. Therefore, pharmacological inhibition of PYK2 effectively dampens intestinal inflammation.

## Discussion

IRF5 has arisen as an attractive therapeutic target for many inflammatory diseases, including IBD. However, its targeting via direct interference with expression or function has proven

difficult so far. In search for kinases that might direct IRF5 activity, we identified PYK2 as a pivotal regulator of IRF5-dependent pro-inflammatory responses. PYK2 is known to mediate signal transduction in cells treated with Toll-like receptor (TLR) ligands[43], although it has never previously been linked to IRF5. Here we show that PYK2 interacts with IRF5 and triggers its phosphorylation, nuclear translocation and binding to dedicated gene promoters. PYK2-deficient cells are impaired in LPS-induced IRF5-driven transcriptional programme.

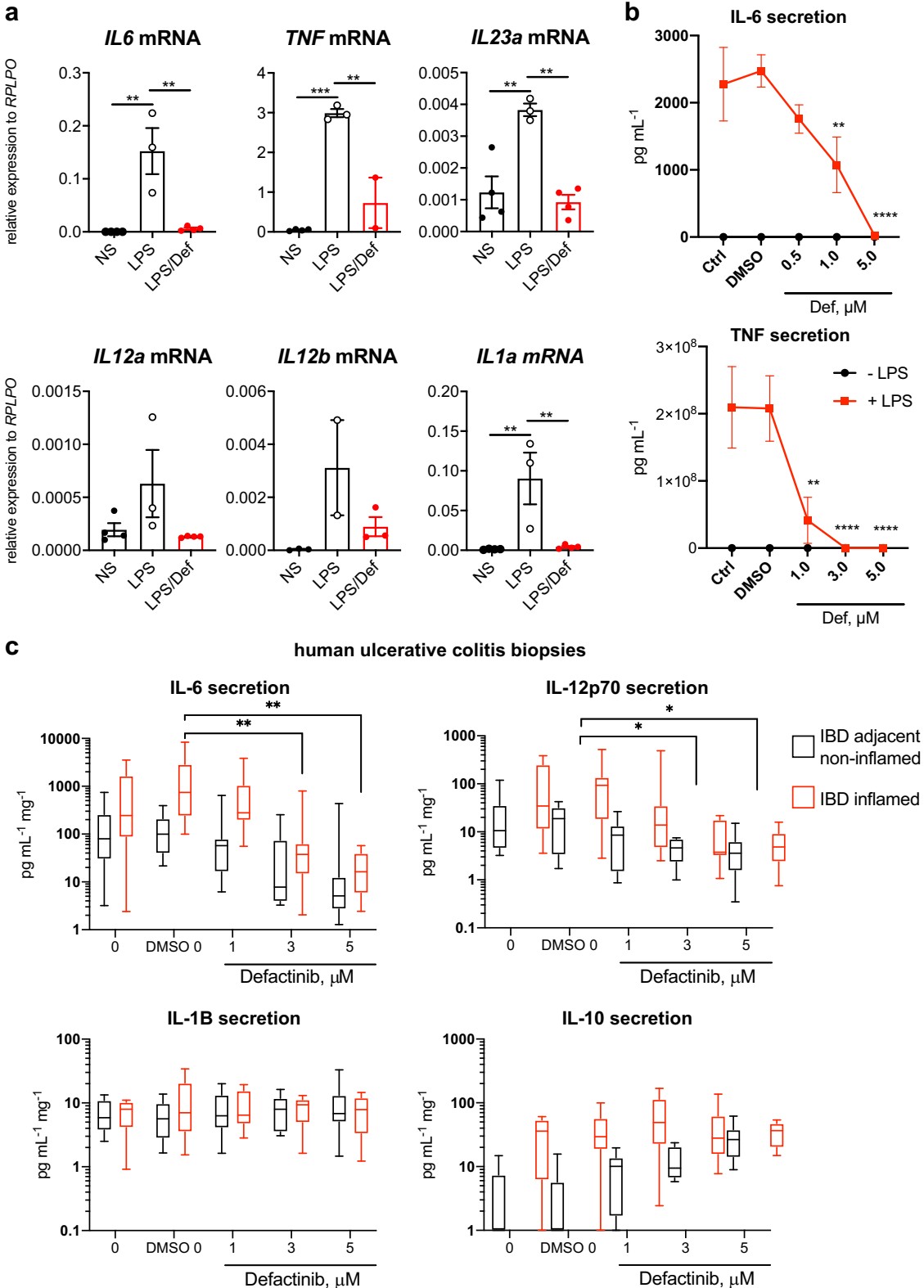

Moreover, pharmacological inhibition of PYK2 with defactinib blocks IRF5 activity in in vitro assays and suppresses LPS-induced pro-inflammatory responses in primary mouse and human monocytes and macrophages. Finally, defactinib reduces pathology in murine colitis models and lowers cytokine production in inflamed colonic biopsies from human patients with ulcerative colitis.

In conclusion, we propose the following pathway involving Pyk2 and IRF5 in macrophages. Upon TLR4 stimulation by LPS or Dectin-1 stimulation by glucan, PYK2 is activated by phosphorylation at Tyr-402[30]. MyD88 is likely to be the essential linking adaptor between TLRs and PYK2/IRF5 complex as earlier studies have shown impairment of PYK2 activation in TLR ligand treated MyD88-deficient cells[30]. PYK2 autophosphorylation has

**Fig. 7 Defactinib inhibits the production of inflammatory mediators by human monocyte-derived macrophages and UC biopsies. a** Cytokine mRNA expression levels in human monocyte-derived macrophages pretreated with defactinib (def, 5 μM) for 1 h followed by LPS stimulation (100 ng/ml) for 2 h. Data were shown as means ± SEM for $n = 3-4$ independent experiments. Statistical significance analysed by one-way ANOVA with Tukey's correction where **$P < 0.01$ and ***$P < 0.001$. **b** Cytokine proteins levels in human monocyte-derived macrophages pretreated with various amounts of defactinib for 1 h, followed by stimulation with LPS (red) for 24 h. Error bars represent mean ± SEM for $n = 3-4$ independent experiments. Statistical significance analysed by two-way ANOVA with Tukey's correction where significant differences relative to control are **$P < 0.01$ and ****$P < 0.001$. **c** Cytokine proteins levels in biopsies from ulcerative colitis patients from non-inflamed and inflamed tissues treated with defactinib at indicated concentrations per mg of tissue. Data were shown as means ± SEM for $n = 10$ human donors and analysed by two-way ANOVA where *$P < 0.05$ and **$P < 0.01$. Box–and- whisker plots represent the median, interquartile range (IQR) and minimum and maximum values. Source data are provided as a Source Data file.

been suggested to occur with the help of Src and possibly other kinases[44,45]. The recruitment of PYK2 to IRF5 upon Dectin-1 stimulation is likely to be Syk-dependent[46]. We show that PYK2 may phosphorylate IRF5 contributing to its activation and transcription of pro-inflammatory cytokines (Supplementary Fig. 9). Multiple sites of phosphorylation detected in IRF5, both serine and tyrosine, highlight the complexity of IRF5 activation and multiplicity of signalling pathways[11,19,47,48].

Although serine phosphorylation has been shown to activate IRF5, little is known about the consequence of tyrosine phosphorylation of IRF5. Tyrosine phosphorylation on IRF5 has been shown in untreated $p53^{-/-}$ tumour cells which disappeared with CPT-11 treatment[49]. Previously, phosphorylation on site Tyr-104 (human/mouse) by BCR-ABL kinase in chronic myeloid leukaemia cells has been linked to negative regulation of IRF5 transcriptional activity[33]. Our functional studies show that Tyr-104 has no effect on IRF5 transactivation. Lyn kinase has also been demonstrated to phosphorylate IRF5 at orthologues of Tyr-312 (human Tyr-329) and Tyr-334 (human Tyr-351) sites, but its kinase activity is dispensable for IRF5 inhibition[19]. Our study indicates that IRF5 is phosphorylated on several tyrosine sites in which Tyr-171 (human Tyr-172) is targeted by PYK2 and may be important for IRF5 activation.

A clear mechanistic interaction between PYK2 and IRF5, in macrophages, identified in this study, combined with an acceptable toxicological profile of PYK2 inhibitor defactinib shown in cancer clinical trials[50], deserves a closer look from the therapeutic perspective. This is further supported by the data from another clinical study showing the efficacy of an inhibitor disrupting the MyD88-TRAF6 complex, which is upstream of PYK2[30], in UC[51].

We propose that defactinib is an attractive molecule for repurposing to treat patients with ulcerative colitis[8,9] and with other inflammatory conditions, such as arthritis[41,52] acute lung injury[41] and atherosclerosis[53], in which IRF5 function in macrophages has been intimately linked to pathogenicity. It may even prove useful in dampening lung inflammation in the severe COVID 19 patients, whose lungs are filled with monocyte-derived macrophages expressing high levels of IRF molecules, including IRF5[54].

## Methods

**Animals**. Mice were bred and maintained under specific pathogen-free (SPF) conditions in accredited animal facilities at the University of Oxford. All procedures were conducted according to the Operations of Animals in Scientific Procedures Act (ASPA) of 1986 and approved by the Kennedy Institute of Rheumatology Ethics Committee. Animals were housed in ventilated cages at a constant temperature (20–23.3 °C) with a 12 h dark/light cycle, with food and water ad libitum. Male C57BL/6 J mice (stock no. 000664) aged 8–11 weeks were purchased from the University of Oxford BMS.

**Plasmids**. Expression constructs encoding full-length IRF5 (v3/v4) and IRF5 truncation mutants with Strep-tag at the N-terminus were previously described[55]. Full-length PYK2 and deletion mutants with Myc tag at the N-terminus were generated by VectorBuilder. Site-specific PYK2 tyrosine mutants with a Flag tag were generated by VectorBuilder. Myc-tagged candidate IRF5 kinases were cloned in the pEAK8-myc vector. The lentiCas9-v2 plasmid for HOXB8 transfections was

a gift from Feng Zhang (Addgene plasmid #52961). Constructs encoding TNF-luciferase, pRL-TK Renilla and pBent empty vector were previously described[55]. The NFkB-luciferase was cloned in the pGL3-Promotor vectors (Promega) and the ISRE-luciferase plasmid (pGL4.45 [luc2P-ISRE-Hygro]) was purchased from Promega.

**Cell culture**. RAW264.7 and 293 TLR4/CD14/MD-2 cells were cultured in DMEM (Lonza) supplemented with 10% FBS (Gibco) and 1% Pen/Strep (Lonza). Bone marrow cells were extracted from wild-type mice and cultured with recombinant GM-CSF (20 ng/mL; Peprotech). On day 8, adherent cells were replated and stimulated with either LPS (100 ng/mL, Enzo) or whole glucan particles (100 μg/mL, Invivogen).

Human monocytes were isolated from leucocyte cones of healthy blood donors. Peripheral blood mononuclear cells (PBMC) were enriched by Ficoll gradient. Monocyte-derived macrophages were generated using adherence method selection and GM-CSF differentiation. Whole PBMC ($50 \times 10^6$) were plated in RPMI-1640 medium for 90 min. After two washes with PBS, adherent monocytes were differentiated into macrophages over 5 days in the presence of 50 ng/mL GM-CSF (Peprotech) in RPMI supplemented with 10% foetal calf serum (FCS) (Sigma-Aldrich), 100 U/mL penicillin, 100 mg/mL streptomycin, 30 mM HEPES and 0.05 mM β-mercaptoethanol.

Hoxb8 macrophage progenitors were a gift from the Sykes Lab (Harvard Medical School). Progenitors were cultured in RMPI-1640 medium (Lonza) supplemented with 10% FBS (Gibco), β-mercaptoethanol (30 mM; Life Technologies), recombinant GM-CSF (10 ng/ml; Peprotec) and β-estradiol (1 μM; Sigma-Aldrich). To differentiate into macrophages, progenitors were washed three times with RPMI-1640 medium to remove the β-estradiol and incubated in complete RPMI-1640 medium supplemented with 10% heat-inactivated FBS, 30 uM β-mercaptoethanol and 20 ng/mL GM-CSF and incubated for 4 days. All cells were incubated in a 5% $CO_2$ humidified atmosphere at 37 °C.

**RNA extraction and quantitative real-time PCR**. Total RNAs were isolated from cells using RNeasy Mini Kit (Qiagen) and reverse transcribed to cDNA using High-Capacity cDNA Reverse Transcription Kit (Life Technologies) as per the manufacturer's protocol. RNA from sorted cells was isolated utilising the RNeasy Micro kit (Qiagen). Real-time PCR reactions were performed on a ViiA7 system (Life Technologies) with Taqman probes (Supplementary Table 1). Gene expression was analysed using the comparative Ct (ΔΔCt) method and normalised against *Hprt* levels or *RPLPO* levels for mouse or humans, respectively.

**RNA-sequencing analysis**. Libraries were sequenced on Illumina HiSeq4000 yielding $>40 \times 10^6$ 150 bp paired-end reads per sample. These were mapped to the mm10 genome using STAR[56] with the options: '–runMode alignReads –outFilterMismatchNmax 2.' Uniquely mapped read pairs were counted over annotated genes using featureCounts[57] with the options: '-T 18 -s 2 -Q 255.' Differential expression was then analysed with DESeq2[58] and genes with fold changes >2 and false discovery rates (FDRs) < 0.05 were deemed to be differentially expressed. Variance stabilised (VST) counts for all DESeq2 differentially expressed genes, likelihood ratio test, false discovery rates (FDRs) < 0.05, were used for dimensionality reduction. For direct comparisons genes with fold changes >2 and FDR <0.05 were deemed to be differentially expressed. Gene set enrichment analysis was performed using one-sided Fisher's exact tests (as implemented in the 'gsfisher' R package https://github.com/sansomlab/gsfisher/).

**Measurement of cytokine production**. Mouse serum cytokine concentrations were analysed by ELISA (Mouse IL12p70, #DY419-05) and Cytometric Bead Array (IL6 #558301, IL1β #560232, BD Biosciences) as per manufacturer's instructions. IL1β, IL6 and IL12p70 concentration in Human biopsy or cell culture supernatants was measured by Cytometric Bead Array (IL1β#558279, IL6# 558276 and IL12p70#558283). IL-10 concentration in human intestinal biopsy supernatants was measured by ELISA (#DY217B-05, R&D systems). TNF was measured in human monocyte-derived macrophage culture supernatants by ELISA (#DY210-05, R&D systems). All cytokine detection was performed according to the manufacturer's instructions.

**Western blots**. Cells were lysed in 1% TX-100 lysis buffer (1% v/v TX-100, 10% v/v glycerol, 1 mM EDTA, 150 mM NaCl, 50 mM Tris pH 7.8) supplemented with protease inhibitor cocktails (Roche). Lysates were incubated on ice for 30 min and cleared by centrifugation at $17,000 \times g$ for 10 min at 4 °C. Protein quantification was performed with the Qubit assay (Thermo Fisher Scientific) according to the manufacturer's protocol. About 10 μg of lysates were boiled in Laemmli sample buffer (Bio-Rad), resolved on a NUPAGE 4–12% Bis-Tris gel (Invitrogen), and transferred onto a PVDF membrane (GE Healthcare) by wet western blotting. Antibodies for western blotting can be found in Supplementary Table 2. Complexes were detected with the chemiluminescent substrate solution ECL (GE Healthcare).

**Subcellular fractionation**. Cell pellets were lysed in cytoplasmic lysis buffer (0.15% NP-40, 10 mM Tris pH 7.5, 150 mM NaCl), incubated on ice for 10 min and layered on top of cold sucrose buffer (10 mM Tris pH 7.5, 150 mM NaCl, 24% w/v sucrose). The lysate was centrifuged at $17,000 \times g$ for 10 min at 4 °C and the supernatant was collected as the cytosolic fraction. The nuclear pellet was lysed in RIPA buffer (150 mM NaCl, 1% NP-40. 0.5% Na-DOC, 0.1% SDS, 50 mM Tris pH 8.0) and sonicated on the Biorupter sonicator (ten cycles of 30 s on/30 s off), followed by centrifugation at $17,000 \times g$ for 5 min at 4 °C. The supernatant was collected as the nuclear fraction.

**Immunoprecipitation**. About $1 \times 10^7$ million cells per immunoprecipitation were seeded and incubated overnight. Media was replaced with serum-free media for 1 h, followed by LPS stimulation at indicated timepoints. Whole-cell extracts were prepared with 1% TX-100 lysis buffer as described above. Lysates were precleared with 100 μL TrueBlot Anti-Rabbit Ig IP beads (eBioscience) by rotating. Samples were incubated with 2 μg antibody for 2 h, followed by 100 μL IP beads (50% slurry) by rotating overnight. Immunoprecipitates were washed three times with IP wash buffer (1% NP-40, 150 mM NaCl, 1 mM EDTA, 20 mM Tris-HCl, pH 8) and eluted by boiling the samples for Laemmli sample buffer (Bio-Rad). Eluates were collected from the beads by centrifugation and resolved on a NUPAGE 4–12% Bis-Tris gel (Invitrogen).

**Generation of Pyk2 and IRF5 CRISPR knockouts**. About 5000 RAW264.7 cells/well were seeded in 96-well plates and infected the next day with PTK2B (ID:MM0000145196), and IRF5 (ID:MM0000200177) lentiviral particles (pLV-U6g-EPCG) provided by Sigma. Cells were transduced at a multiplicity of infection (MOI) of 10 in medium containing polybrene (8 μg mL$^{-1}$) and spun at $1500 \times g$ for 1 h. After overnight incubation, media was replaced with fresh media, and selected with 4 μg mL$^{-1}$ puromycin (InvivoGen) for 2 weeks.

Hoxb8 macrophage progenitors were transduced with lentiCas9-v2 lentivirus targeting exon 2 of Irf5 (ID: ENSMUSG00000029771, gRNA ACCCTGGCGCCA TGCCACGAGG) and exon3 of PYK2 (ID: ENSMUSG00000059456, gRNA CCCT ATTCGCCCACTCAGG). Briefly, the lentiCas9-v2 lentivirus were produced from HEK-293FT cells transfected with the lentiCas9-v2 plasmid mixed at a 2:1:1 DNA ratio of the lentiviral packaging plasmids pMD2.G (Addgene plasmid #12259) and psPAX2 (Addgene plasmid #12260) at a 2:1:1 ratio. Media was replaced 16 h post-transfection. Two days post transfection, the lentivirus containing mediums were harvested, filtered and added onto Hoxb8 macrophage progenitor cells at a final concentration of 8 ug/ml polybrene. Transduced cells were allowed to grow for additional 4 days and selected with 6 ug/ml Puromycin for the targeted knockout of Irf5 and Pyk2.

**Chromatin Immunoprecipitation**. About $1 \times 10^7$ million cells per ChIP were seeded and incubated overnight. GM-BMDMs cells were pretreated with defactinib or DMSO vehicle for 1 h, followed by LPS (100 ng/ml) for 2 h. RAW264.7 cells were stimulated with LPS (500 ng/ml) for 2 h. Cells were fixed in formaldehyde, quenched with Tris pH 7.5 and washed in PBS. After centrifugation, cells were lysed for 10 min in LB1 buffer (50 mM Hepes-KOH, pH 7.5, 140 mM NaCl, 1 mM EDTA, 0.5% NP-40, 10% glycerol, 0.25% Triton X-100) supplemented with proteases inhibitors. Nuclei were pelleted at $1350 \times g$ and resuspended in LB2 buffer (10 mM Tris-HCl, pH 8.0, 200 mM NaCl, 1 mM EDTA, 0.5 mM EGTA) for 10 min. Nuclei were pelleted at $1350 \times g$ and resuspended in LB3 buffer (10 mM Tris-HCl, pH 8.0, 100 mM NaCl, 1 mM EDTA, 0.5 mM EGTA, 0.1% Na-deoxycholate, 0.5% Na-Lauroylsarcosine). Samples were sonicated with a Bioruptor (Diagenode) for eight cycles (GM-BMDMs) or ten cycles (RAW264.7). Lysates were immunoprecipitated with 5 μg of anti-IRF5 (ab21689; Abcam), anti-RNA Polymerase II (MMS-128P; Biolegend) or Rabbit Anti-Mouse IgG (ab46540; Abcam). Immunoprecipitated DNA was purified with the PCR Purification Kit (Qiagen). qPCR analysis was carried out in duplicates and represented as % input. Primer sequences can be found in Supplementary Table 1.

**Kinase inhibitors screening and luciferase reporter assay**. RAW264.7 cells were seeded in eighteen 96-well plates at 50,000 cells/well a day before transfection as described above. One hour prior to LPS treatment, cells were treated with 10 uM of inhibitors in quadruplicates. For experiment wells ($n = 4$ for each inhibitor set AG-AK, total amounts for 16 plates (AG-AJ1-4)—80 wells per plate and two plates (AK1,2-3,4)– 96 wells per plate) 21 ml of Opti-Mem (Gibco) was mixed with DNA: 50 μg of pBent-HA-IRF5, 50 μg of pGL3-5′TNF-luc and 25 μg of pEAK8-Renilla,

vectors described in refs. [24,59]. About 5 ml of Opti-Mem mixed with 200 μl of Plus reagent was added to the DNA solution and incubated for 5–15 min. Then, 5 ml of Opti-Mem was mixed with 500 μl of Lipofectamine LTX reagent, added to the DNA/Plus solution and incubated for 30 min. For controls (amount for four plates —two 4 well-rows each), 800 μl of Opti-Mem was mixed with DNA: 2 μg of pBent-HA-IRF5 or pBent2-empty, 2 μg of pGL3-5′3′-TNF-luc and 1 μg of pEAK8-Renilla. About 200 μl of Opti-Mem mixed with 8 μl of Plus reagent was added to the DNA solution and incubated for 5–15 min. Then, 200 μl of Opti-Mem was mixed with 20 μl of Lipofectamine LTX reagent, added to the DNA/Plus solution and incubated for 30 min. To transfect cells, 20 μl of the DNA/transfection reagent mix was added per well. Next day (after ~16–18 h), cells were pre-incubated for 1 h with 20 μl of inhibitor (or 1% DMSO to control wells) in serum-free DMEM (final conc. 0.1, 1 or 10 μM in 1% DMSO). Then, 1 μg/ml of LPS was added to the cells and 6 h later the culture medium was and the plate-bound cells were kept frozen (−20 °C). Cells were lysed using the Dual-Glo Luciferase Assay kit (Promega) according to the manufacturer's protocol and analysed in a FLUOstar Omega microplate reader (BMG Labtech). Raw firefly luciferase activities (or values normalised against Renilla luciferase activities) in wells incubated with the kinase inhibitors were divided by the luciferase activity values in the control wells (DMSO vehicle only, cells expressing IRF5 and stimulated with LPS) and expressed as part of a whole or a percentage of IRF5 reporter activity (which was 1 or 100% in cells treated with DMSO only).

**Kinase assays**. 293 ET cells were plated at 250,000 cells/well in six-well plates and a day later were transfected with 1 μg of pBent2-HA-IRF5 and 1 μg of plasmid encoding one of the myc-tagged candidate IRF5 kinases using Lipofectamine2000™ (Life technologies) according to the manufacturer's protocol. The cell lysates were subjected to kinase assays using a modification of an established protocol[60]. Cells were washed in PBS and lysed on the ice in kinase reaction buffer (20 mM HEPES pH 7.5, 137 mM NaCl, 0.5 mM EGTA, 25 mM MgCl2, 0.2% Triton X-100, 10% Glycerol) with added protease (EDTA-free! complete-mini protease inhibitor cocktail, Roche, #11836170001) and phosphatase inhibitors (phosphatase inhibitor cocktail II, Sigma, #P5726). About 2 mM of TCEP (#C4706, Sigma), 1 mM of GTP (#G8877, Sigma) and 50 μM S-γ-ATP (ab138911, Abcam) was added sequentially to lysing samples. The reactions were mixed by vortexing and kept a rocking surface for 1 h at 37 °C. The reactions were stopped by adding 50 mM EDTA and moving them on ice. p-Nitrobenzyl mesylate (PNBM, ab138910, Abcam) was dissolved in DMSO to 50 mM. PNBM working solution was prepared by adding 5 μL of deionized water five times mixing after each addition to 25 μL of the PNBM stock, and then was added 1/10 to kinase reactions (at 2.5 mM), which were further incubated for 2 h at room temperature. The bulk of the reactions were subjected to immunoprecipitations using anti-thiophosphate-ester antibody (1 μg per reaction, ab92570, Abcam) and Protein G Sepharose 4 Fast Flow Media (#11524935, GE Healthcare) to pull down phosphorylated IRF5. The rest of the reactions and the pull-downs were mixed with SDS PAGE loading buffer and subjected to SDS PAGE following Western blotting.

**Mass spectrometry analysis**. IRF5 was immunoprecipitated from LPS-stimulated WT and PYK2 KO RAW264.7 cells ($5 \times 10^7$ cells) as described in the immunoprecipitation section. Eluents were subjected to in-solution digestion as described previously[61]. In brief, samples were reduced and alkylated before double precipitation with Chloroform/Methanol as described in ref. [62]. Protein pellets were resuspended in 50 μL 6 M urea for solubilisation. The samples were diluted to 1 M Urea in 100 mM Tris buffer for tryptic digest. Following overnight digestion, peptides were acidified with 3% Formic acid and desalted with solid-phase extraction Sola cartridges (Thermo). Peptides were eluted with 600 uL glycolic acid solution (1 M glycolic acid, 80% acetonitrile, 5% trifluoroacetic acid). Phospho-peptide enrichment was performed using a TiO$_2$ protocol as described in ref. [63] with eluates from the Sola cartridges adjusted to 1 mL with 1 M glycolic acid solution and incubated for 5 min with 50 uL TiO$_2$ bead slurry solution. Bead washes (200 uL) were carried out as previously described. In short, beads were sequentially washed with 200 uL glycolic acid solution, ammonium acetate solution (100 mM ammonium acetate in 25% acetonitrile) and 10% acetonitrile solution repeated in triplicate. Phospho-peptides were eluted, following incubation for 5 min at room temperature with 50 ul ammonia solution (5%) and centrifuged, this was repeated in triplicate. The three eluate fractions were combined and dried using a SpeedVac and pellets were stored at −80 °C until analysis. For analysis by nano-liquid chromatography-tandem mass spectrometry (nLC-MS/MS), a Dionex UHPLC system coupled to an Orbitrap Fusion Lumos mass spectrometer was used as described previously[64]. Raw MS files were subjected to processing using PEAKS (version 8.5) software and searched against the UniProtSP (P56477) Mus Musculus database. Searches included the data refine, denovo PEAKS and PEAKS PTM modes, the latter of which included phosphorylation on Ser (S), Thr (T) and Tyr (Y) residues.

***Helicobacter hepaticus*-induced colitis model**. Male mice aged 8–11 weeks were free of known intestinal pathogens and negative for Helicobacter species. Animals from each experimental group were cohoused. On days 0, 1 and 2, mice were injected i.v. with 1 mg/kg defactinib or vehicle (5% DMSO, 2.5% Solutol HS

(Sigma), 2.5% absolute ethanol, 90% Dulbecco's PBS). Daily, starting from day 3, mice were injected i.p. with 5 mg/kg defactinib or vehicle. 30 min after the initial i.v. injection, mice were infected with $1 \times 10^8$ colony forming units Hh on days 0 and 1 by oral gavage with a 22 G curved, blunted needle (Popper & Sons). Mice were injected intraperitoneally once on day 0 with 1 mg anti-IL10R blocking antibody (clone 1B1.2). Infected mice were monitored daily for colitis symptoms. Mice were euthanized with $CO_2$ at d7 post infection, and organs were harvested for analysis. Experimental groups consisted of 5–7 co-housed mice and were blinded to the researchers.

**Isolation of lamina propria leucocytes.** Colons and/or caeca were harvested from mice, washed in PBS/BSA and content flushed with forceps. Intestines were then opened longitudinally and washed once more before blotting to remove mucus. Gut tissue was then cut into 1 cm long pieces and placed in a 50 mL centrifuge tube (Greiner) in ice-cold PBS + 0.1% BSA. Colons were incubated two times at $300 \times g$ in 40 mL HBSS + 0.1% BSA + 1% Penicillin-Streptomycin (PS, Lonza) + 5 mM EDTA (Sigma-Aldrich) at 37 °C for 10 min before the supernatant was aspirated. Tissue was placed in 40 mL PBS + 0.1% BSA + 1% PS for 5 min. Intestines were then incubated with 20 mL RPMI + 10% FCS + 1% PS + 2.5 U/mL Collagenase VIII (Sigma-Aldrich) + 2 U/mL DNAse I (Roche), shaking for 45 mins–1 h at 37 °C. Supernatant was filtered through a 70-μm-cell strainer to which 30 mL of ice-cold PBS + 0.1% BSA + 1% PS + 5 mM EDTA was added to ablate collagenase/DNase activity. Cells were washed in 30 mL PBS/BSA before filtering once more through a 40-μm-cell strainer. The cells were then pelleted by centrifugation at 400 rcf for 10 min at 4 °C and resuspended in 1 mL RPMI + 10% FCS + 1% PS before counting.

**Flow cytometry.** CBA quantification of cytokine levels were performed on a FACSCanto II (BD) and analysed using Flowjo (Treestar Inc.). Acquisition of mouse samples was performed using either LSR II or Fortessa X20 flow cytometers with FACSDiva (BD), followed by analysis in Flowjo (Treestar Inc.). Gating strategy in Fig. S7d.

**Extracellular labelling of cells.** About $5 \times 10^5$–$2 \times 10^6$ cells were plated on U-bottom 96-well plates. Cells were washed twice with 150 μL FACS buffer (PBS + 0.1% BSA + 1 mM EDTA + 0.01% Sodium Azide) at 400 rcf for 3 min 4 °C. Cells were then Fc blocked for 10 min with αCD16/CD32 (BD) 1/100 in 20 μL FACS buffer at room temperature (RT) followed by washing once in 150 μL FACS buffer. Fixable Viability Dye eFluor780 (ThermoFisher) and primary extracellular antibodies (Supplementary Table 1) were added for 30 min at 4 °C in 20 μL FACS buffer in the dark. Labelled cells were then washed twice with 150 μL FACS buffer. Cells were then fixed for 30 min in 50 μL Cytofix (BD), washed twice with 150 μL FACS buffer and resuspended in 200 μL FACS buffer before acquisition.

**FACS sorting.** Colon lamina propria cells were prepared as described above. A small aliquot of each sample was stored in RNAlater (Sigma-Aldrich) for further processing. Two to three samples were pooled in order to gain sufficient numbers for sorting. The cells were labelled as described above with the antibodies in Supplementary Table 3, except no fixation step was performed. Labelled cells were washed twice with 1 mL FACS buffer and resuspended in 500 uL FACS buffer containing DNAse I (10 ug/mL, Roche). Sorting of the cells was performed into 500 uL RNAlater on FACSAria III (BD Biosciences) at the Kennedy Institute of Rheumatology FACS facility.

**Culture of UC patient colonic mucosal biopsies.** Intestinal pinch biopsies were obtained from Ulcerative Colitis patients registered in the Oxford IBD Cohort, attending the John Radcliffe Hospital Gastroenterology Unit (Oxford, UK). This cohort comprises 1896 patients with UC, median age 31 at diagnosis, treated with biological therapy (23%) or conventional steroids/immunomodulators (77%) for active disease, in addition to mesalazine. Biopsies were collected during routine endoscopy. Informed, written consent was obtained from all donors. Human experimental protocols were approved by the NHS Research Ethics System (Reference numbers: 16/YH/0247). Biopsies were washed in PBS and transferred into wells containing RPMI-1640 + 10% FCS + 20 μg/mL G418 (Thermo Fisher) + 20 U/mL Pen/Strep and cultured for 24 h.

**UC biopsy viability assessment.** Biopsies were fixed in 4% PFA in PBS (#30525-89-4, Santa Cruz) for 24 h at RT and transferred to 70% ethanol. Fixed biopsies were then dehydrated and embedded in paraffin blocks, and 5 μm sections were cut. Embedding and sectioning of tissues was carried out by the Kennedy Institute of Rheumatology Histology Facility (University of Oxford). The viability of intestinal biopsies was measured by TACS TdT in situ (Fluorescein) TUNEL assay (#4812-30-K, R&D systems) according to the manufacturer's instructions. Sections were then mounted in Glycerol Mounting Medium with DAPI and DABCO (#ab188804, Abcam) and cover-slipped. Images of three non-sequential sections per sample were acquired. Three images per section were acquired at 20x magnification using a BX51 microscope (Olympus). To generate the apoptotic index,

the total cell number was enumerated by counting DAPI$^+$ and TUNEL(FITC)$^+$ cells in ImageJ, and calculating the percentage of total cells that were TUNEL$^+$.

**Histopathological assessment.** Following euthanasia, 0.5 cm pieces of caecum, and proximal, mid and distal colon were fixed in PBS + 4% paraformaldehyde (Sigma-Aldrich). Fixed tissue was embedded in paraffin blocks, and sectioned using a microtome and stained with Haematoxylin and Eosin (H&E) by the Kennedy Institute of Rheumatology Histology Facility (Kennedy Institute of Rheumatology, University of Oxford). Sections were scored in a blinded manner by two researchers according to ref. [65].

**Cell viability.** About 50,000 cells/well were seeded and incubated overnight. Cell viability was assessed using the Promega CellTiter-Glo Luminescent kit per the manufacturer's protocol and luminescence was measured in a FLUOstar Omega microplate reader (BMG Labtech) on the MARS Data analysis software. Samples were tested in triplicate and normalised to untreated wells.

**Protein isolation from colon tissue.** About 1.5 ml Bioruptor Microtubes were filled with 250 mg of Protein Extraction Beads (Diagenode) and filled with RIPA buffer (supplemented with protease and phosphatase inhibitors). Ten micrograms of tissue were added to the tubes and vortexed briefly. Tubes were sonicated on the Bioruptor Pico with 30 s ON/30 s OFF for six cycles at 4 °C. After every two cycles, tubes were vortexed. The supernatant was transferred to a new tube and centrifuged at $17,000 \times g$ for 10 min at 4 °C. The supernatant was transferred to a new tube and 80 μg of lysate was used for immunoblot analysis.

**Reporting Summary.** Further information on research design is available in the Nature Research Reporting Summary linked to this article.

## Data availability
The RNA-sequencing data generated in this study have been deposited in the GEO database under accession code GSE141082. The proteomics data and MS raw files have been deposited to Proteome Xchange Consortium via the PRIDE[66] partner repository with the dataset identifier PXD014033. Source data are provided with this paper.

## Code availability
All code used in the current study has been stored at https://github.com/Tariq-K?tab=repositories.

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

## Acknowledgements

We are grateful to Dr. Jelena Bezbradica-Mircovic (University of Oxford) for their critical reading of the manuscript and useful comments. This work was supported by the Versus Arthritis (Ph.D. studentship 209966 to H.A.), the BRC3 Gastroenterology and Mucosal Immunology (A.L.C. and S.P.L.T.), the Kennedy Trust for Rheumatology Research (PhD studentship AZT00040 to D.L.B.), the Chinese Science Council (PhD studentship to Z.A.) and the Wellcome Trust (Investigator Award 209422/Z/17/Z to I.A.U.: I.A.U., H.L.E. and K.Z.). Work in the BMK lab was funded by the Chinese Academy of Medical Sciences (CAMS) Innovation Fund for Medical Science (CIFMS), China (grant number: 2018-I2M-2-002). We thank GSK for providing their published kinase inhibitor library for this project and Dr David B Sykes (Harvard Medical School) for sharing HoxB8-GM-CSF cells and culture protocols with us.

## Author contributions

G.R. and H.A. performed the majority of wet-lab experiments. A.L.C., D.L.B. and H.L.E. performed in vivo experiments. A.L.C. carried out work on human samples. S.Bullers, C.P., Z.A. and K.Z. contributed to the histological and imaging analysis of colon sections; S.Bonham, R.F. and B.M.K. carried out mass spectrometry experiments and data analysis. T.K. performed RNA-seq analysis. L.J.-D. and S.P.L.T. helped edit the manuscript. G.R., H.A. and I.A.U. wrote the manuscript.

## Competing interests

The authors declare no competing interests.
