## [Peer Review File · Nature Communications]

REVIEWER COMMENTS

Reviewer #4 (Gut inflammation, cytokine signaling) (Remarks to the Author):

The revised manuscript is improved compared to the original submission.

Reviewer #5 (Bacterial immunity, TLR signaling) (Remarks to the Author):

Ryzhakov, Almuttaqi et al report a tremendous amount of data that aims to show that PYK2 controls intestinal inflammation via activation of IRF5 in macrophages.

This work was submitted to Nature and seen by 4 primary reviewers, who were not overly impressed and raised a number of very sensible questions, some of which were adequately dealt with by the authors, and others not very adequately.

My own general impression is that this work is both overdocumented and underdocumented, and is almost impossible to read and digest for an average reader. The authors need to choose what is their question, and focus on it.

Take the title: it literally says that PYK2 controls intestinal inflammation via activation of IRF5 in macrophages. If that is the conclusion, they need to compare intestinal inflammation Pyk2 wt and KO animals, if possible in macrophage-conditional mutants; better still if the PYK2 KO is not a null but a kinase-dead one. This was not done, and the authors argue that they could not get the Pyk2 KOs. They rather follow the rather convoluted road of finding a PYK2 inhibitor, and testing it in a model of intestinal inflammation.

They come down to two kinase inhibitors, whose specificity was not well known, and they argue that they are PYK2-specific. Most of the paper is about this. My own title would be "Defactinib inhibits PYK2 phosphorylation of IRF5 and reduces intestinal inflammation".

The connection to genetic studies is weak and should not be mentioned in the abstract because no direct experiment on the risk variants was done here.

The overdetailed description of the screening for inhibitors of that are active in the cell-based assay of TNF expression is near-useless, also because the authors focus on PYK2 while other targets might be as interesting. Their choice is mostly personal, so it does not need to be based on too many data, which can go to supplementary anyway.

The absolute specificity for PYK2 is a false problem: they just need to show that IRF5 phosphorylation is involved and, if blocked, blocks inflammation. The identification of the phosphorylation site of IRF5, though, is important. Here, the first objection of reviewer 1 is critical. The authors undermine their own conclusion that Tyr171 phosphorylation controls IRF5 activity, since the impact of the mutation to Phe is minimal (rather surprisingly, they do not compare Y172F to wt in fig 3c).

Whether PYK2 is acting via TRIF or MyD88 is not central at this time.

We all know that overexpression systems can give false positives, and that's why confirming the conclusion in vivo is essential. I do not think everything should be tested in vivo, though, so the request to test the PYK2:IRF5 interaction in primary macrophages looks excessive. The experiments in HOXD8 macrophages seem adequate.

Reviewer #6 (Innate immunity, NFkB signaling) (Remarks to the Author):

(Only private remarks to the editor)

REVIEWER COMMENTS

Reviewer #4 (Gut inflammation, cytokine signaling) (Remarks to the Author):

The revised manuscript is improved compared to the original submission.

Reviewer #5 (Bacterial immunity, TLR signaling) (Remarks to the Author):

Ryzhakov, Almuttaqi et al report a tremendous amount of data that aims to show that PYK2 controls intestinal inflammation via activation of IRF5 in macrophages.

This work was submitted to Nature and seen by 4 primary reviewers, who were not overly impressed and raised a number of very sensible questions, some of which were adequately dealt with by the authors, and others not very adequately.

My own general impression is that this work is both overdocumented and underdocumented, and is almost impossible to read and digest for an average reader. The authors need to choose what is their question, and focus on it.

Take the title: it literally says that PYK2 controls intestinal inflammation via activation of IRF5 in macrophages. If that is the conclusion, they need to compare intestinal inflammation Pyk2 wt and KO animals, if possible in macrophage-conditional mutants; better still if the PYK2 KO is not a null but a kinase-dead one. This was not done, and the authors argue that they could not get the Pyk2 KOs.

They rather follow the rather convoluted road of finding a PYK2 inhibitor, and testing it in a model of intestinal inflammation.

They come down to two kinase inhibitors, whose specificity was not well known, and they argue that they are PYK2-specific. Most of the paper is about this. My own title would be "Defactinib inhibits PYK2 phosphorylation of IRF5 and reduces intestinal inflammation".

As explained in previous round of revision, we could not obtain PYK2 KO mice, which are not publicly available. Defactinib is an inhibitor of PYK2, developed by Pfizer and validated by Oxford Structural genomics consortium, which is in clinical trials for cancer. We have amended the title.

The connection to genetic studies is weak and should not be mentioned in the abstract because no direct experiment on the risk variants was done here.

Our aim was to emphasise a possible connection of two genetic risk factors for IBD, which so far had not been reported for any other IBD genetic factors. We have removed the sentences on the proposed connection between the two genetic risk factors from the

abstract. We left references to IRF5 and PYK2 being proposed IBD genetic risk factors in the introduction, as these are published studies.

The overdetailed description of the screening for inhibitors of that are active in the cell-based assay of TNF expression is near-useless, also because the authors focus on PYK2 while other targets might be as interesting. Their choice is mostly personal, so it does not need to be based on too many data, which can go to supplementary anyway.

The overdetailed description was partly a result of responding to reviewers' questions from the first round of Nature revision. We have streamlined it back.

We would not have discovered Pyk2 without an initial screen of inhibitors. Thus, we believe this is an important starting point for presenting this work. As shown in Sup Fig 1, Pyk2 was the only kinase that passed all functional and prioritisation tests. While we have placed most of the data related to the technicalities of the screen in Supplementary Figure 1, we think that the results of the screen and the kinases identified warrant the main figure, as they can lay the basis for further investigation by other groups.

The absolute specificity for PYK2 is a false problem: they just need to show that IRF5 phosphorylation is involved and, if blocked, blocks inflammation.

We believe that this link has already been demonstrated in our system of PYK2 deficient RAW264 macrophages. We have now separated analyses achieved in this system (Figs 2, 3) from the data obtained using over-expression systems (Fig 1).

Specifically: **endogenous** IRF5 activation (measured by ChIP) is blocked in PYK2 KO macrophages (Fig 2d). In the same cells **endogenous** inflammatory gene expression is reduced (Fig 2e). This is also validated in HoxB8 macrophages (Fig 2f). There is no good IRF5-phospho abs to measure IRF5 phosphorylation. The mass spec data indicate that **endogenous** IRF5 phosphorylation at Y171 is no longer detected in PYK2 KO cells (Fig 3a,b).

The identification of the phosphorylation site of IRF5, though, is important. Here, the first objection of reviewer 1 is critical. The authors undermine their own conclusion that Tyr171 phosphorylation controls IRF5 activity, since the impact of the mutation to Phe is minimal (rather surprisingly, they do not compare Y172F to wt in fig 3c).

Comparison of Y172F to wt was presented in the original submission. During the revision we changed it to accommodate previous reviewers' comments. We have reverted to that presentation (now Sup Fig S4d). We have tuned down the conclusions and now only present phosphorylation data in Fig 3c, which show the reduction in the level of PYK2-

mediated phosphorylation for IRF5 Y172F mutants. The significant but not dominant reduction of luciferase-reporter activity in the presence of the same mutants is now shown in Sup Fig S4.

Whether PYK2 is acting via TRIF or MyD88 is not central at this time.

We agree, again it was done in response to previous reviewers. The text has been revised to streamline the message and irrelevant to the main story points have been removed.

We all know that overexpression systems can give false positives, and that's why confirming the conclusion in vivo is essential. I do not think everything should be tested in vivo, though, so the request test the PYK2:IRF5 interaction in primary macrophages looks excessive. The experiments in HOXD8 macrophages seem adequate.

We are pleased that the reviewer accepts our HoxB8 experiments as adequate.

Reviewer #6 (Innate immunity, NFkB signaling) (Remarks to the Author):

(Only private remarks to the editor) referee #6 felt that previous points 1/3/4/5/6 from referee #1, and previous points 2/3/5 from referee #2, have not been satisfactorily addressed.

Without knowing what exactly the reviewer 6 said, it is hard for us to respond. When addressing the points of previous reviewers, we felt we went as far as we could in relation to the story line and with the currently available tools.

There is a considerable difference in opinion between the six reviewers involved, i.e. reviewer 5 thinks that the point 2 made by reviewer 2 on Myd88 and TRIF pathways, and highlighted by reviewer 6, is irrelevant.

REVIEWERS' COMMENTS

Reviewer #5 (Remarks to the Author):

The authors have significantly refocused and streamlined their manuscript, and have convincingly identified the IRF5 tyrosine phosphorylated by PYK2.

I also see that they have accepted my suggestion for the title.

I am fully satisfied with their work.

Reviewer #6 (Remarks to the Author):

I am satisfied with the revisions and think the paper is now acceptable for publication